https://doi.org/10.1038/s41467-022-30744-3　　**OPEN**

# Berry curvature-induced local spin polarisation in gated graphene/WTe$_2$ heterostructures

Lukas Powalla [1,7], Jonas Kiemle[2,3,7], Elio J. König[1], Andreas P. Schnyder [1], Johannes Knolle [3,4,5], Klaus Kern [1,6], Alexander Holleitner[2,3], Christoph Kastl [2,3] & Marko Burghard [1✉]

Experimental control of local spin-charge interconversion is of primary interest for spintronics. Van der Waals (vdW) heterostructures combining graphene with a strongly spin-orbit coupled two-dimensional (2D) material enable such functionality by design. Electric spin valve experiments have thus far provided global information on such devices, while leaving the local interplay between symmetry breaking, charge flow across the heterointerface and aspects of topology unexplored. Here, we probe the gate-tunable local spin polarisation in current-driven graphene/WTe$_2$ heterostructures through magneto-optical Kerr microscopy. Even for a nominal in-plane transport, substantial out-of-plane spin accumulation is induced by a corresponding out-of-plane current flow. We present a theoretical model which fully explains the gate- and bias-dependent onset and spatial distribution of the intense Kerr signal as a result of a non-linear anomalous Hall effect in the heterostructure, which is enabled by its reduced point group symmetry. Our findings unravel the potential of 2D heterostructure engineering for harnessing topological phenomena for spintronics, and constitute an important step toward nanoscale, electrical spin control.

[1] Max-Planck-Institut für Festkörperforschung, Heisenbergstrasse 1, D-70569 Stuttgart, Germany. [2] Walter Schottky Institut and Physics Department, Technical University of Munich, Am Coulombwall 4a, D-85748 Garching, Germany. [3] MCQST, Schellingstrasse 4, D-80799 Munich, Germany. [4] Department of Physics TQM, Technical University of Munich, James-Frank-Strasse 1, D-85748 Garching, Germany. [5] Faculty of Natural Sciences, Department of Physics, Imperial College London, London SW7 2AZ, UK. [6] Institut de Physique, Ecole Polytechnique Fédérale de Lausanne, CH-1015 Lausanne, Switzerland. [7] These authors contributed equally: Lukas Powalla, Jonas Kiemle. ✉email: m.burghard@fkf.mpg.de

The field of spintronics has experienced a major leap forward by the recent advent of high-quality devices comprised of layered two-dimensional (2D) materials[1,2]. This advancement is enabled by several unique advantages of the 2D materials platform. First, the vast library of van der Waals (vdW) materials readily provides several candidates for efficiently generating current-driven spin polarisations, such as topological insulators with their spin-momentum-locked surface states or Weyl semimetals with their spin-polarised Fermi arc surface states[3–5]. Moreover, atomically thin, layered magnets exhibit pronounced magnetic anisotropy and the possibility to control the magnetism by electric fields, thus enabling new spin generation or read-out schemes[6]. Combining such individual components into vdW heterostructures[7] allows to further tailor the spintronic functionality through spin-orbit coupling (SOC), proximity exchange, or spin-orbit torques[8–12]. VdW stacks also appear to be close-to-ideal platforms for the implementation of magnetisation switching through symmetry, in analogy to recent experiments on thin film heterostructures[13]. Ultimately, by complementing interface design with external control via gate fields[14], the electronic properties can be manipulated on-demand, enabling for example switchable spin textures or topological band properties. Such type of devices are for example promising to emulate the functions of neurons and synapses, potentially paving the way for low-power logic and data processing beyond CMOS electronics[15,16].

Gate-tunable spin-charge interconversion as a cornerstone of spin-electronics can be achieved by combining graphene and a strong SOC 2D material serving as a spin transport channel and as spin generator, respectively[2,10]. This functionality has been demonstrated experimentally for vdW heterostructures incorporating a 3D topological insulator[17], a trivial or topological 2D semimetal[18–20], or a 2D semiconductor[21–25]. Generally, the electrically detected spin-signals can persist up to room temperature, which is highly desirable for applications[17,18,20–22,25,26]. Despite this progress, however, another relevant aspect of vdW-based heterostructures, namely their topology-related spintronic potential, most prominently via intrinsic Berry curvature (BC)-mediated mechanisms, has remained largely unexplored. BC imparts an effective magnetic field in momentum space, while deforming the electron motion in real space, thus leading to exotic transport properties like various Hall effects[27]. Moreover, the dipole component of the BC becomes relevant in the presence of inversion asymmetry, which can be exploited for, e.g., non-linear optoelectronic transport[28]. Importantly, this may be associated with substantial, local spin polarisation effects, whose investigation requires suitable read-out techniques of high spatial resolution[29].

Here, we use Kerr rotation (KR) microscopy to detect, with sub-micrometre resolution, possible signatures of current-induced spin polarisation in graphene/WTe$_2$ heterostructures as a function of gate voltage. In these devices, the graphene component allows for current injection into the adjacent WTe$_2$ in a manner not achievable by conventional metal contacts. Our theoretical analysis, taking into account the independently determined charge current flow in the heterostructure, reveals that the inherent symmetry breaking due to the presence of graphene at the heterointerface is crucial for explaining the observed phenomena.

## Results

### Basic device characterisation

The device in Fig. 1a consists of an ~20 nm thick WTe$_2$ ribbon interfaced to a single-layer graphene stripe beneath, such that the two long axes form an angle close to 90°. The long (short) axis of the WTe$_2$ crystal corresponds to its

$a$-axis ($b$-axis), as confirmed by polarisation-resolved Raman spectroscopy (Supplementary Fig. S1). The $c$-axis points out-of-plane. The overlay in Fig. 1a is a current-induced KR map, with the red and blue areas representing KR signals of opposite sign. Generally, KR spectroscopy detects the off-diagonal elements of the refractive index, i.e. the difference in the refractive indices for circularly left-handed and right-handed photons, via a corresponding polarisation rotation of the reflected light. The polar configuration (normal incidence) used in our experiments probes the in-plane Hall tensor, which is typically associated with a corresponding out-of-plane magnetisation (or spin polarisation) as discussed later in detail. While KR microscopy has thus far mainly been used to detect magnetic phase transitions in 2D materials under equilibrium conditions, we investigate here the KR signal generated under electrical current flow in a vdW heterostructure. For the current-induced KR microscopy, an AC current at frequency $\omega$ (on the order of kHz) is passed along the graphene stripe between the contacts labelled 1 and 3 (i.e., parallel to the $b$-axis of WTe$_2$), all other contacts are floating. The gate voltage $V_g$ is applied to the silicon substrate with respect to contact 3, which serves as the voltage reference (ground). All measurements were conducted at a bath temperature of 4.2 K. The current-induced out-of-plane spin polarisation (aligned with the $c$-axis) is then locally resolved via the KR angle $\theta_K^\omega$ detected at the fundamental frequency $\omega$ of the alternating bias (Fig. 1b). The sign (or direction) of the KR is then related to the sign (or direction) of the local magnetisation. It should be noted that the present KR signals cannot be accounted for by the non-linear anomalous Hall effect (AHE) previously reported for WTe$_2$[30,31]. In the latter case, crystal symmetry is reduced due to the quasi-2D character of the few-layer sheets, which leads to a current-induced out-of-plane magnetisation perpendicular to the charge current direction[32].

In Fig. 1c, transfer curves of the graphene stripe within the above device are shown for increasing AC bias up to $V_{13}^\omega = 1.5$ V. At low AC bias amplitudes, two resistance maxima corresponding to two Dirac points are visible, pointing towards the presence of graphene regions with different doping levels[33]. The Dirac point at gate voltage $V_g \approx 0$ V can be assigned to the bare graphene leads from the metallic contacts to the junction. The second Dirac point at $V_g \approx 20$ V indicates moderate $p$-doping and is attributable to the graphene section proximitized by WTe$_2$. With increasing AC bias amplitude, the resistance maximum associated with the Dirac point at the heterojunction progressively decreases and finally vanishes. The latter suggests a change of the current path: at large bias, the adjacent WTe$_2$ opens a parallel current path and shunts the proximitized graphene, with a vertical current flow from graphene to the WTe$_2$ upon entering the junction region (and analogously backwards to graphene close to the opposite junction edge).

### Local heat dissipation and current flow

To further corroborate such a non-trivial current distribution in our heterostructure, we utilised our polarisation sensitive Kerr detection scheme. We applied an alternating source-drain voltage with a frequency of $\omega = 3.33$ kHz and monitored the polarisation rotation of the reflected light at the second harmonic $2\omega$ of the source-drain frequency by a lock-in measurement. Previous experiments on metals have shown that the polarisation rotation $\theta_K^{2\omega}$ at modulation frequency $2\omega$ can capture the effect of Joule heating on the optical reflectivity[34] and, therefore, the spatial dependence of heat dissipation due to the local currents and resistances in the device structure. For WTe$_2$, a $2\omega$ modulation can be understood by the temperature-dependent variation of the refractive indices along the different crystal axes, i.e., it probes the birefringence.

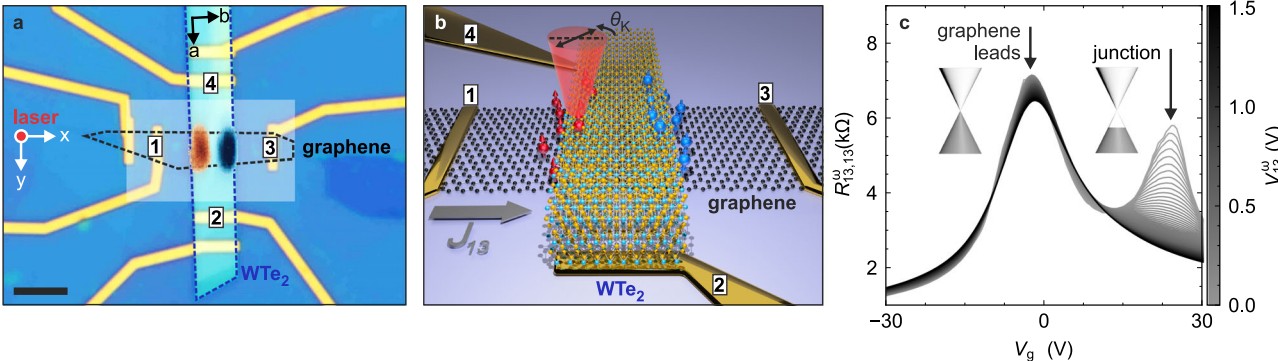

**Fig. 1 Optical KR microscopy and graphene transfer curves. a** Optical microscope image of a heterostructure comprised of graphene (black dashed line), WTe₂ (blue dashed line) and hBN capping on a back-gated Si/SiO₂ substrate. The WTe₂ crystallographic axes are indicated. The overlay shows the current-induced KR signal at the junction using a colour code as in Fig. 3, for a current applied between the contacts labelled 1 and 3. Scale bar, 5 μm. The probed area is marked by the shaded rectangle. **b** An oscillating current $j_{13}$ at frequency $\omega$ is applied along the graphene stripe. At the heterojunction, the current-induced out-of-plane spin component (red and blue arrows) is locally read-out by a polar KR measurement at frequency $\omega$. The effect of Joule heating is detectable at twice the oscillation frequency $2\omega$. **c** Two-terminal resistance $R^{\omega}_{13,13}$ of the graphene stripe measured between contacts 1 and 3 of the device in **a** using an AC voltage $V^{\omega}_{13}$ that increases stepwise up to 1.5 V. The gate voltage $V_g$ was applied to the silicon back contact. The two Dirac points reveal different doping levels (see inset) for the bare graphene leads (charge neutral) and the proximitized graphene ($p$-doped).

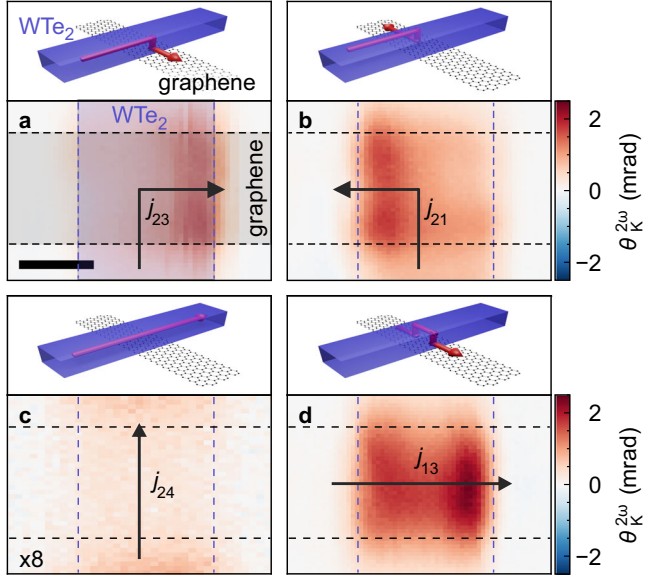

**Fig. 2 Mapping of local charge current.** Current-induced polarisation rotation $\theta^{2\omega}_K$ (lower panel) for different bias configurations of the graphene/WTe₂ junction (upper panel). **a**, **b** Bias applied between graphene and WTe₂. Bias applied along (**c**) WTe₂ and (**d**) graphene only. In **c**, the colour scale was expanded by a factor of eight for better visibility. The arrows indicate the direction of the ac current $j$ as expected from the bias configuration. The numbers indicate the contacts used for biasing. All other contacts are floating. The scale bar is 2 μm. Experimental parameters are $V_g = 0$ V, **a** $V_{23} = 3$ V, **b** $V_{21} = 3$ V, **c** $V_{24} = 4$ V, **d** $V_{13} = 1.5$ V.

Figure 2 shows spatial maps of $\theta^{2\omega}_K$ around the WTe₂/graphene heterointerface, which were obtained in four different bias configurations. For source-drain bias applied vertically across the junction, the maximum signal is always located along the direction of current flow, i.e., either to the right (contacts 2 and 3 are connected, Fig. 2a) or the left (contacts 2 and 1 are connected, Fig. 2b) towards the corresponding electrical contact on graphene. The signals scale approximately quadratically with the applied AC current amplitude, as expected for Joule heating (Supplementary Fig. S2). Thus, we conclude that the polarisation rotation demodulated at $2\omega$ indeed reflects the local dissipation

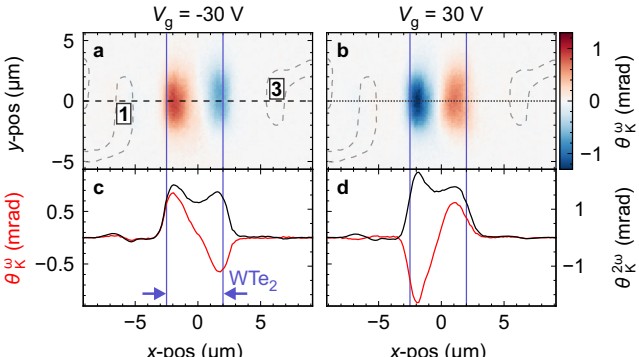

**Fig. 3 Gate-dependent KR microscopy.** Spatially-resolved KR, detected on the device in Fig. 1 under AC current injection along the graphene stripe ($V_{13} = 4$ V) for **a** $V_g = -30$ V and **b** $V_g = 30$ V. Grey dashed lines highlight the metal electrodes, labelled 1 and 3. **c**, **d** Profiles of the current-induced Kerr angle $\theta^{\omega}_K$ and the Joule heating-induced polarisation rotation $\theta^{2\omega}_K$, along the horizontal dotted/dashed lines in **a** and **b**. Vertical solid lines indicate the WTe₂ ribbon.

and hence the local charge current density. On this basis, the signal maxima in Fig. 2a, b can be explained by a local vertical current flow between the two layers. In comparison, for in-plane biasing along the WTe₂ channel (contacts 2 and 4 are connected, Fig. 2c), a much weaker signal occurs at the central interface, indicative of homogeneous Joule heating within the less resistive WTe₂. Finally, when the bias is applied along the proximitized graphene, which is nominally in-plane, we detect an increased optical response from the edges (contacts 1 and 3 are connected, Fig. 2d). These two pronounced $2\omega$ signals indicate the local out-of-plane current flow from graphene to WTe₂ and vice versa.

**Gate-dependent Kerr microscopy.** Figure 3 displays current-induced KR maps of the above device for two gate voltage regimes. The bias current is applied along the graphene stripe between the contacts labelled 1 and 3 (cf. Fig. 1). Very similar behaviour has been observed on two other devices (Supplementary Figs. S3 and S4). At $V_g = 30$ V ($n$-type regime), strong KR of opposite sign appears around the two edges of the junction area (Fig. 3a). Changing the gate voltage to $V_g = -30$ V ($p$-type regime), reverses the KR polarity while its magnitude, location

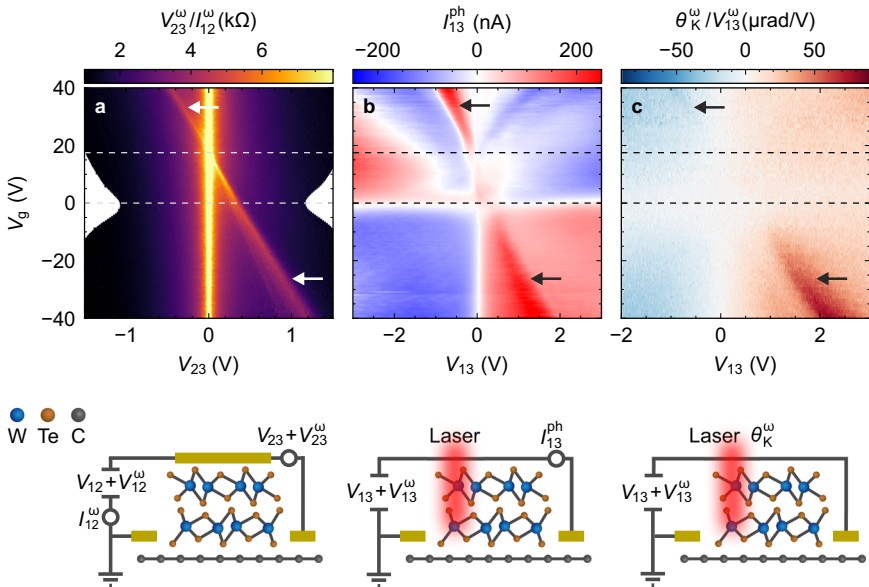

**Fig. 4 Electronic and optical interface spectroscopy. a** Differential resistance $V_{23}^{\omega}/I_{12}^{\omega}$ as a function of gate voltage $V_g$ and DC drain-source voltage $V_{23}$. A vertical tunnel barrier is assumed due to the vdW gap between graphene and WTe$_2$. **b** Local photocurrent acquired at the graphene/WTe$_2$ junction as a function of $V_g$ and drain-source voltage $V_{13}$. **c** Differential KR signal $\theta_K^{\omega}/V_{13}^{\omega}$ vs. $V_g$. In each case, the corresponding experimental configuration is displayed at the bottom. The horizontal dotted lines indicate the positions of the two Dirac points. The horizontal white or black arrows in the upper panels mark the Dirac point of the proximitized graphene. All data were taken at 4.2 K on a second heterostructure device.

and spatial extent are barely affected (Fig. 3b). The location of the maximum KR coincides with the Joule heating-induced $2\omega$ signal, underscoring the connection between KR signal and current flow through the heterostructure. As elaborated later by our theoretical model, the sign change of the KR signals between Fig. 3a, b can be traced back to the sign reversal of current injection into WTe$_2$, depending on whether the underlying graphene is $n$- or $p$-doped. Current is injected into WTe$_2$ by momentum conserving tunnelling of electrons, while microscopically, the momentum of conduction (valence) electrons in graphene is antiparallel (parallel) to the current.

**Tunnelling spectroscopy.** To shine further light on the microscopic correlation between vertical charge currents and the Kerr signal, we performed interlayer transport spectroscopy (Fig. 4a) of a second device (Supplementary Fig. S3). With the source-drain bias applied vertically across the junction, the graphene and WTe$_2$ can be regarded as planar tunnelling electrodes, possibly due to the weak coupling across the vdW gap[35]. We added a small AC voltage onto the DC bias to measure the differential resistance $V_{23}^{\omega}/I_{12}^{\omega}$ across the heterointerface (for details see "Methods"). The bias- and gate voltage-dependent tunnelling resistance (Fig. 4a) features a prominent peak that linearly shifts along the diagonal from the upper left to the bottom right. It can be assigned to the Dirac point of the proximitized graphene. The diagonal shift arises due to the interplay of the applied gate voltage, which dopes the graphene, and the bias voltage, which compensates the difference between the WTe$_2$ Fermi level and the proximitized Dirac point.

Intriguingly, this proximitized Dirac point appears also in a nominally lateral transport configuration with the bias current injected along the graphene stripe (contacts 1 and 3). This is apparent from Fig. 4b, displaying the local photocurrent $I_{13}^{ph}$ for an excitation at the junction (laser wavelength = 800 nm), and Fig. 4c which shows the differential KR signal $\theta_K^{\omega}/V_{13}^{\omega}$. In both maps, there is an onset at a bias voltage corresponding to the Dirac point of the proximitized graphene, akin to Fig. 4a. It is

noteworthy that the linear, gate-independent background in the Kerr spectroscopy (Fig. 4c) is due to Joule heating. Overall, the gate- and bias-dependencies are consistent with the theoretical picture below, which relies on minority carriers in graphene.

## Discussion

One plausible explanation for the current-induced KR signals in Fig. 3 could be a linear spin Hall effect, driven by an electrical current within the WTe$_2$ ribbon. However, this effect can be ruled out, as the electrical current direction, spin current direction, and spin direction must be mutually orthogonal[36], such that it is not possible to account for the observed KR signal at the left and right edge of the junction (cf. Fig. 1a), neither for vertical (along the $c$-axis of WTe$_2$) nor horizontal charge current flow (along the $b$-axis of WTe$_2$) within the ribbon. Consistently, our control experiments in bare WTe$_2$ do not show detectable KR by any in-plane currents (Supplementary Fig. S5). An alternative mechanism could be charge-to-spin conversion induced by lateral charge transport within the proximitized graphene with an in-plane spin component due to a Rashba-like interaction, as well as an out-of-plane spin texture due to a valley-Zeeman like interaction[20]. Again, this scenario cannot explain our observations, as it would be expected to lead to a Kerr signal that is homogeneously distributed along the graphene/WTe$_2$ interface, rather than being localised at the interface edges as observed by experiment.

While the above two mechanisms fail to account for the observed KR signals, we argue that a local, current-induced magnetisation originating from an AHE[37] is responsible for the observed phenomena. Generally, the current density in response to the fast optical probe field $E_j(\nu)$ is:

$$j_i(\nu) = \sigma_{ij}(\nu)E_j(\nu). \qquad (1)$$

The antisymmetric components of the conductivity tensor $\sigma_{ij}(\nu)$ determine the Kerr response. Linear antisymmetric conductivity components in the unbiased system vanish due to time reversal symmetry. As the latter is effectively broken in our measurement configuration due to the slow bias field $E_l(\omega)$, we

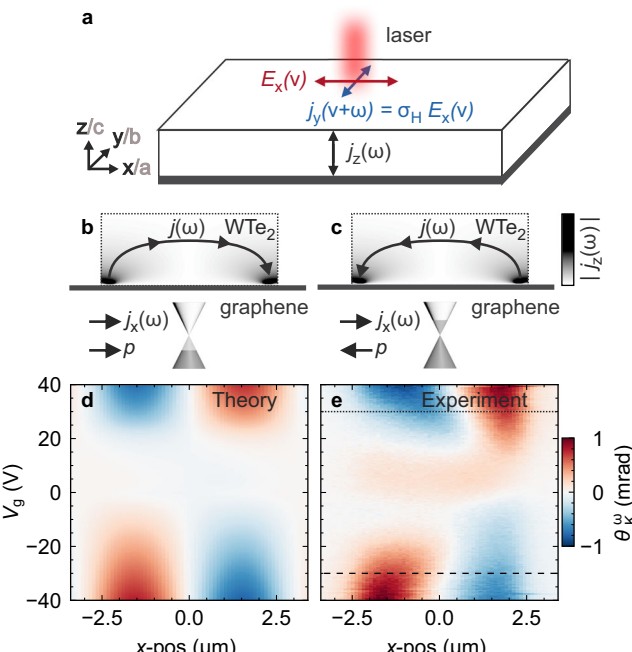

**Fig. 5 Theory of current-induced KR in WTe₂. a** Schematic drawing of the electric fields associated with the linearly polarised laser light (red), the induced non-linear Hall response (blue), and the vertical component of the ac current flow (black). **b**, **c** Sketch of the current flow profiles in the WTe₂, shown for **b** $p$-type and **c** $n$-type doping of the proximitized graphene under overall current flow $j_x$ in positive $x$-direction; $p$ is the carrier momentum. **d** Theoretically calculated and **e** experimentally measured $\theta_K^\omega$ ($V_{13} = 3\,\text{V}$) as a function of $x$ and $V_g$. The dashed/dotted lines indicate where the spatial profiles in Fig. 3 were taken. The data are from the device in Fig. 1.

consider the non-linear antisymmetrised current response:

$$j_i(\nu + \omega) = \epsilon_{ijk}\lambda_{lk}(\nu, \omega)E_l(\omega)E_j(\nu). \tag{2}$$

From this equation, it follows that the combined actions of the fast optical probe field and the slow bias field lead to a non-linear current response tensor $\lambda_{lk}$. Equation (2) is a generalisation of the non-linear Hall response and circular photogalvanic effect at $\nu = -\omega$[28,30–32,38–42]. By definition, this current is always transverse to the optical field $E_j(\nu)$. If the current direction is also perpendicular to the direction of light propagation, this will result in a Kerr angle in the reflected field. Figure 5a sketches the relevant electric fields in our measurement. The laser light is incident along $z$ with polarisation along $x$ and correspondingly the Hall current is along $y$. With this specific choice of coordinates we define the Hall conductivity $\sigma_H(\omega) = (\sigma_{xy}(\omega) - \sigma_{yx}(\omega))/2 = \lambda_{lz}E_l(\omega)$ which is responsible for the Kerr angle $\theta_K(\omega) \propto -\text{Re}(\sigma_H(\omega))$[43].

According to the theory of the AHE[37], $\sigma_H$ is proportional to the magnetisation $\langle S_z \rangle$ and thus effectively measured by the Kerr angle. Further to this, recent works have shown that $\lambda_{ji}$ can be expressed in terms of the BC dipole tensor $D_{ji}^{(\Omega)} = \langle \Omega_i v_j \rangle_{\text{FS}}$, i.e., the Fermi surface average of the BC $\Omega_j(\mathbf{p})$ and velocity $v_i$[32,39].

Crucially, non-centrosymmetric materials allow for a finite BC dipole $D_{ji}^{(\Omega)}$[32] and $\sigma_H(\omega) \propto D_{jz}^{(\Omega)}j_j(\omega)$ can be thus induced by a charge current $j_j(\omega)$ inside the topological metal, leading to (see Supplementary Information for details):

$$\langle S_z \rangle(\omega) \propto \theta_K(\omega) \propto D_{jz}^{(\Omega)}j_j(\omega). \tag{3}$$

This establishes a relationship between Kerr microscopy, spintronics, topological band theory, and non-reciprocal

transport coefficients[34,44,45], which also determine the non-linear AHE[30,31,41,42] and photocurrents in topological metals[28,38,40]. Consistently with Eq. (3), our control measurements show that the KR angle depends linearly on the amplitude of the quasi-DC bias field (Supplementary Fig. S2), beyond a gate-dependent onset voltage as discussed below. Moreover, the Kerr angle does not depend on the amplitude of the optical probe field (Supplementary Fig. S6), ruling out any higher-order effects[46] as well as local photocurrents to be the origin of the Kerr response and corresponding spin accumulation.

While the current-induced anomalous Hall response in the bulk of either graphene or bulk WTe₂ vanishes due to crystalline symmetries, controllable net spin-charge interconversion is enabled in the graphene/WTe₂ heterostructures due to their reduced interface symmetry.

The symmetry group of $T_d$-WTe₂ is $P_{mn2_1}$, and it contains a 180° screw rotation about the $c$-axis as well as a $b \rightarrow -b$ mirror symmetry. The latter is broken due to strain and the relative misalignment of the crystalline axes of WTe₂ and graphene. This imposes $\lambda_{xz} = \lambda_{yz} = 0$, while $\lambda_{zz} \neq 0$ becomes symmetry allowed resulting in $j_y(\nu + \omega) = -\lambda_{zz}(\nu, \omega)E_z(\omega)E_x(\nu)$ (compare Eq. (2)).

To connect the above homogeneous Kerr response to the local current density, we model the diffusion currents in the heterostructure taking into account the experimentally observed interlayer current. Before that, we outline the modifications to diffusion theory in topological materials. From standard diffusion theory, the density $n(\mathbf{x}, t)$ and the regular current density $j_i(\mathbf{x}, t)$ obey:

$$j_i = -D\partial_{x_i}n \qquad \text{(Fick's 1st law),} \tag{4a}$$

$$\partial_t n = D\sum_i \partial_{x_i}^2 n \qquad \text{(Fick's 2nd law).} \tag{4b}$$

For simplicity, the diffusion coefficient $D$ is kept isotropic. For a Berry-curved material, we derive an additional differential equation for the BC density $\varpi_i(\mathbf{x}, t)$ (see Supplementary Information):

$$\varpi_i = -\tau D_{ji}^{(\Omega)}\partial_{x_j}n \qquad \text{(topological extension).} \tag{4c}$$

The total current $j_i^{(\text{tot})}$ is given by the regular current and the BC contribution:

$$j_i^{(\text{tot})} = j_i + \epsilon_{ijk}\varpi_j eE_k. \tag{5}$$

The second term stems from the anomalous velocity contribution to the semiclassical equations of motion of Bloch electrons and $E_k$ the local electric field. The local in-plane Hall response is then given by $\sigma_H(\mathbf{x}) = -\int_0^h dz\varpi_z(\mathbf{x})$.

We now address the mechanism of current injection into the WTe₂ slab due to charge carrier hopping across the heterointerface in the presence of a current $j_0$ in the graphene sheet. We consider momentum conserving tunnelling, such that the injected current comprises the imbalance current $\partial_x[n_e + n_h]$ rather than a charge current $\partial_x[n_e - n_h]$ where $n_e, n_h$ are electron and hole densities in graphene. This follows from the fact that the velocity (i.e., current) of the same momentum in $n$- and $p$-doped graphene is opposite (Fig. 5b, c).

Using the continuity equation, current injection thus enters effectively as a source term in Eq. (4b), $D\nabla^2 n = \partial_x[\bar{D}\partial_x[n_e + n_h]]\delta(z)$, where $\bar{D}$ is a constant. Furthermore, as the source term is given by the derivative of the imbalance current, it is dominated by its spatial dependence. As opposed to a transport current, the latter is determined by the relaxation profile of minority carriers. These are massively generated when the bias voltage across the proximitized region exceeds a threshold value set by the distance to the Dirac point and qualitatively explains the gate-dependent minimal bias for the signal onset in Fig. 4c. Fitting this model to experimental data (Supplementary Fig. S7), yields a calculated KR response (Fig. 5d)

within a single band approximation which reproduces both the onset and the sign change of the measured KR (Fig. 5e). We note that, in the gate voltage regime between the two Dirac points, the KR signal, although being rather weak, has non-zero spatial average and hence cannot be interpreted based on $\bar{\omega}_z = -\tau D_{zz}^{(\Omega)} \partial_z n \propto j_z$, because the total current going into and out of the $WTe_2$ must vanish by Kirchhoff's law. We hypothesise that the weak KR stems from a higher-order non-linear current-field relation beyond Eq. (2).

The BC dipole-induced Kerr signal in our heterostructure is surprisingly large. In general, the magneto-optical detection of current-induced spins in simple metals is challenging due to the small ratio of optically probed to spin-polarised electrons[47], the short spin relaxation time[48], and their low optical activity compared to semiconductors. Nonetheless, this has been accomplished with the aid of current-modulation techniques[34,49]. The present KR angles $\theta_K^\omega$ in the mrad range are approximately six orders of magnitude larger than for metal wires under ac current injection[34], although the current densities are comparable (on the order of $1 \times 7\,A\,cm^{-2}$). We assign the large KR directly to the pronounced symmetry breaking of the heterointerface in conjunction with the significant interlayer currents. It should furthermore be noted that even in a lock-in detection scheme, it is difficult to completely eliminate spurious contributions of heating effects to the detected first harmonic signal, in close correspondence to previous Kerr microscopy studies of metallic systems[34,47,48]. Nonetheless, that the current-induced spin polarisation makes a significant contribution to the Kerr signal is underscored by angular-dependent measurements of the Kerr signal (Supplementary Fig. S8).

In principle, the experimentally observed Kerr angles can be a direct measure of the BC dipole, which is a Fermi surface property of the electronic system in the low temperature limit. This requires a proper description of the optical conductivities and the carrier scattering times entering the Boltzmann equation. At optical frequencies, provided that the crystal is sufficiently clean and the bands are well resolved, extrinsic mechanisms related to disorder scattering become negligible, while the intrinsic BC related mechanisms are dominant[39]. Indeed, for resonant optical interband excitation in 2D semiconductors, such as $MoS_2$, simple two-band models have been successfully applied to describe a strain-induced BC dipole and a corresponding valley magnetisation[50]. Here, for the particular case of semimetallic $WTe_2$, the situation at optical frequencies is more complicated, because many possible interband transitions exist[51]. From this point of view a THz to far-IR read-out would be appealing. In this range, the response of the electronic system is dominated by the intraband Drude terms and for $\omega\tau \gg 1$ the non-linear Hall conductivity, and also the Kerr effect, becomes independent of the scattering time and a direct probe of the BC dipole[32,39]. However, this would be achieved at the cost of sacrificing spatial resolution, for the case of an optical far-field read-out.

Finally, we would like to underscore the great potential of imaging the local heat dissipation in the 2D heterostructures by applying the same polarisation sensitive detection scheme at different harmonics of the excitation frequency. Such imaging provides highly valuable insights into the local current flow and, in the present study, revealed an unexpected interlayer exchange current flow even for nominally in-plane transport in the graphene/WTe₂ heterostructure. For utilising 2D heterostructures in spintronic applications, the demonstrated knowledge of the local charge as well as spin current distribution is crucial. Considering observed spin diffusion lengths on the order of $10\,\mu m$ in bare graphene[10] and $2\,\mu m$ in $MoTe_2$[3], far-field optical probes can indeed access physically and technologically relevant length scales via local and non-local optoelectronic read-out schemes[40,52]. Our generic model devised for the current-induced spin polarisation

merges classical diffusion theory with topology and it should be applicable to other 2D materials and their heterostructures where BC plays a crucial role, such as transition metal dichalcogenides or nodal line semimetals[41,53].

In summary, using magneto-optic Kerr microscopy we have demonstrated that a nominal in-plane electrical current flowing in a graphene/WTe₂ heterostructure induces a pronounced nonequilibrium spin density with out-of-plane orientation in the WTe₂ layer. The spin polarisation profile is asymmetric along the current injection direction and depends on the dominant type of charge carriers. It is attributed to an out-of-plane current flowing from graphene to WTe₂, with the tunnelling electrons experiencing the non-zero BC dipole of the heterostructure, enabled by breaking of centrosymmetry at the heterointerface. Our theoretical model traces back the observed Kerr signals to a non-linear Hall effect in the heterostructure, combined with a topologically-modified diffusion theory. Overall, our measurements provide a valuable basis for implementing tunable topological electronics and local control of spin polarisation in 2D vdW heterostructures, and complement electrical detection schemes for in-plane spins towards probing out-of-plane spins in vdW heterostructures.

## Methods

**Fabrication of graphene/WTe₂ heterostructure devices**. The samples were prepared using an all-dry viscoelastic transfer method. Single-layer graphene was mechanically exfoliated from natural graphite flakes (NGS Trading & Consulting GmbH, Germany) onto a Si/SiO₂ wafer. Subsequently, bulk crystals of WTe₂ (hq graphene, The Netherlands) were mechanically cleaved and exfoliated onto a PDMS stamp. Crystals with high aspect ratios, indicative of preferential cleavage along the *a*- and *b*-crystal axes, and a thickness of 15–25 nm were identified by optical contrast and transferred from the stamp onto the graphene monolayer. All steps were performed under ambient.

**Magneto-optic Kerr measurements**. The magneto-optic Kerr measurements were carried out using a confocal dip-stick microscope with a sample bath temperature of 4.2 K (Supplementary Fig. S9). A linearly polarised cw-laser at $\lambda_{laser} = 800$ nm was focused onto the sample with a diffraction-limited spot size of ~800 nm and a laser power of 50 μW. The reflected beam was guided through a 50:50 beamsplitter, a half-wave plate, a Wollaston beamsplitter, and detected by an amplified balanced photodetector. An alternating bias current with a frequency of $\omega = 3.33$ kHz was applied to the sample, the polarisation change detected by the balanced photodetectors was read-out using a lock-in amplifier simultaneously at the fundamental ($\omega$) and the second harmonic ($2\omega$) frequency. Spatially-resolved scanning was performed by moving the sample using a *xy*-piezo scanner.

**Tunnelling and local photocurrent measurements**. The vertical charge transport experiments on the graphene/WTe₂ heterojunction were carried out using a gate-dependent four-probe differential conductivity measurement at $T_{bath} = 4.2$ K using standard lock-in detection. A DC voltage, to which a small AC amplitude (1 mV at 77 Hz) was added, was applied to the graphene/WTe₂ junction contacts. The resulting ac voltage was measured at the opposing graphene/WTe₂ contacts. The gate voltage was applied to the Si-back gate using a source/measure unit.

## Data availability

All data needed to evaluate the conclusions in the paper are present in the paper and the Supplementary Information.

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

## Acknowledgements

Experimental work at TUM was supported by Deutsche Forschungsgemeinschaft (DFG) through the German Excellence Strategy via the Munich Center for Quantum Science and Technology (MCQST)—EXC-2111-390814868, SPP-2244 "2D Materials—Physics of van der Waals [hetero]structures" via Grant KA 5418/1-1, HO 3324/12-1, HO 3324/13-1, and the excellence cluster "e-conversion". C.K. acknowledges support through TUM International Graduate School of Science and Engineering (IGSSE). Experimental work at MPI was supported by Deutsche Forschungsgemeinschaft (DFG) through SPP-2244 "2D Materials—Physics of van der Waals [hetero]structures" via Grant BU 1125/12-1 and the DFG Grant "Weyl fermion-based spin current generation" BU 1125/11-1. We acknowledge technical support by T. Reindl, A. Güth, U. Waizmann, M. Hagel and J. Weis from the Nanostructuring Lab of the Max Planck Institute for Solid State Research.

## Author contributions

J.Ki. and L.P. contributed equally and are both first authors. M.B., A.H., and C.K. conceived and designed the experiments. L.P. fabricated the heterostructures. J.Ki. and L.P. performed the optoelectronic measurements. J.Ki., L.P., and C.K. analysed the data. E.J.K. developed the theoretical analysis and wrote the theory section with input from A.P.S. and J.K. All authors co-wrote and reviewed the manuscript.

## Funding

## Competing interests

The authors declare no competing interests.
