## [Peer Review File · Nature Communications]

Reviewers' Comments:

Reviewer #1:

Remarks to the Author:

Comments

This manuscript describes the Kerr microscope measurements with a theoretical explanation of current-induced spin polarization at the heterostructure interface consisting of graphene and van der Waals WTe_2 . The results and analyses are intriguing and may warrant publication in Nature Communications, but the presentations and descriptions are not detailed enough to correctly understand the importance and significance of their results. Once the readability problems pointed below are resolved, the manuscript could be acceptable.

Firstly, Figures 1a,b, and c, together with captions, are not informative enough to understand the experimental details; the authors should explain how they applied the gate voltage V_g by using an equivalent circuit diagram as in Fig. 3. They should define V_{dc} and R_{2p} and show how their values were experimentally determined in the main text or the method section. The authors should describe it as supplementary information if space does not permit it.

Secondly, about theoretical description from Eqs 1 to 3c, it is essential to make the equations more easily understandable for broad readers, the physical meaning of variables such as ε_{ijk} in the equation on line 194, n and τ in Eq. 3 (Why does the Berry curvature density depend on the density n ?)

Reviewer #2:

Remarks to the Author:

This work concerns the measurement of a current-induced spin polarization in a WTe₂/Graphene heterostructure and its relation to the Berry curvature. This is a subject of relevance to understand the generation and transport of spins in 2D heterostructures. The measurements as well as the theoretical results appear to be of high quality. However, the main message of the paper is hard to grasp. Moreover, the theoretical proceedings and some of the measurements should be better explained to be accessible to a broad audience.

The main experimental point is the demonstration that an in-plane current flowing in graphene induces a (strong?) spin accumulation in a WTe₂ layer on top of graphene, as measured by the magneto-optic Kerr effect. The spin polarization is out-of-plane and depends on the type of charge carriers dominating transport in graphene. This observation is attributed to a diffusive out-of-plane current flowing from graphene to WTe₂ and to the nonzero Berry curvature dipole of the noncentrosymmetric WTe₂ layer. Furthermore, the Kerr effect is related to the nonlinear Hall effect of WTe₂.

After reading the paper, one still does not understand if the Kerr effect measurements provide a quantitative measurement of the Berry curvature dipole, nonlinear Hall conductivity, and spin accumulation in WTe₂. Whereas it can be argued that the Kerr angle provides spatially-resolved signatures of these effects, their quantitative evaluation relies on so many assumptions and parameters that only order of magnitude estimates can be given. Indeed, the fits according to Eq. S18 are used only to confirm the qualitative agreement of the model with the experimental results. The authors should clarify what the scope and limitations of the measurements are.

The generation of an out-of-plane spin accumulation can in principle be assigned to different mechanisms depending on the actual direction of current flow in the heterostructure. Also, the spin texture of graphene in proximity to WTe₂ is known to acquire out-of-plane components. The authors should explain more clearly if and why the out-of-plane spin polarization is an exclusive signature of the nonlinear Hall effect. They should also specify more clearly how the WTe₂ is electrically connected for the experiments reported in Fig. 1.

Is there any relationship between the laser polarization, crystal axes of WTe₂, and the spin-polarization direction? Changing the orientation of the WTe₂ layer relative to the current flow and/or the linear polarization direction of the laser could lend further support to the authors' analysis.

The discussion of the nonlinear Hall conductance is not straightforward. The nonlinear Hall conductance is related to the product of two electric field components, which in transport measurements reduce to the square of the electric field driving the current (see, e.g., ref. 22). However, in this case there is no electric field applied in the z-direction, only the diffusion current arising from the carriers' concentration gradients. Perhaps the authors imply that the electric field E_j is the one that builds up as a result of the diffusion process, but this is not explained. Moreover, the second component of the electric field is the one due to the incident light at frequency ν . The interplay of the different types of electric fields in determining the Hall conductivity is not at all evident to readers who have no notion of optical measurements. The nonlinear Hall effect and the different types of electric fields should be introduced in a more pedagogical way. Presenting a schematic diagram showing the different field/current directions and the spin polarization direction allowed by symmetry would help. Moreover, the authors should discuss how the Kerr rotation depends on the current amplitude in graphene (data in Fig. S3a) and on the laser intensity in order to support their model.

The discussion also lacks a broader perspective relating these qualitative findings to other measurements of charge-spin interconversion in 2D heterostructures. What do these results add to the experiments revealing the spin Hall effect and inverse spin galvanic effect in graphene/transition metal dichalcogenide systems?

Aside from these points, I suggest to add a top view of the wiring diagrams for both Fig. 1 and 2 and a clearer explanation of what is actually measured. What is R_{2p} in Fig. 1c, a two-point resistance vs V_g ? The tunneling spectroscopy is $\Delta V / \Delta I$; does ΔV measure an ac voltage and $V_{ds, meas}$ a dc voltage?

In the Supplementary Information, the theoretical section is hard to follow, at least for a person that does not share the same background as the one who wrote it. I would first expect a discussion of how the nonlinear Hall effect emerges from the topological Fick theory. This would allow the reader to identify at first sight which quantities need to be modeled in order to understand the measurements. Instead, the nonlinear Hall conductivity is given almost for granted on line 138 based on results presented in Refs. 17 and 32. Moreover, the role of the imbalance current could be introduced in a more intuitive way. Finally, in connecting the Kerr rotation to the Hall conductivity (Eq. S17), the Berry curvature dipole is replaced by the Berry curvature Ω and then dropped, which brings back the question of how useful are the measurements to determine topological parameters. Several quantities that appear in this equation have not been defined (v_w , v_w , W , l , ...). p_F is likely the Fermi momentum in units of $1/\text{length}$, but this is not written. Overall, the notation is rather involved and sometimes confusing. For example, the Berry curvature dipole and diffusion constants share the same symbol D .

In conclusion, this paper presents interesting measurements and theoretical modelling of nonlinear effects in graphene/WTe2 heterostructures. However, the interpretation of the data as well as the modelling of the results are not explained with sufficient clarity. This makes it hard to evaluate the conclusions and impact of this work. I suggest that the authors resubmit their manuscript after an extensive revision that improves on the points described above.

Reviewer #3:

Remarks to the Author:

In this work, Powalla et al. report the measurement of the current-induced spin density at the interface between WTe2 and graphene monolayer probed by magneto optical Kerr microscopy. The authors observe a nonequilibrium spin density polarised normal to the plane of the heterostructure, that displays an antisymmetric profile along the injection direction, and whose magnitude can be controlled by a gate voltage. The authors interpret their experimental results as a consequence of the Berry curvature dipole experienced by the carriers tunnelling from graphene to WTe2.

This is an interesting result that will certainly attract significant attention in the community. An interesting idea is the connection between Berry curvature dipole, nonlinear anomalous Hall effect and nonequilibrium spin density. However, whereas the qualitative connection between S_z and the nonlinear Hall effect and its associated Berry dipole D_{zz} is rather reasonable, I find the formal relationship is still not well established. The manuscript would gain in clarity by further developing this connection.

- As a matter of fact, WTe2 is known to display nonlinear anomalous Hall effect, which has been theoretically associated with the Berry curvature dipole. In Eq. (1), the authors establish a qualitative connection with the S_z component of the spin density and the Berry curvature dipole. It would be more convincing to provide the explicit expression that relates these two quantities rather than only providing (unjustified) proportionality relations. What is the physics connecting S_z and the nonlinear Hall effect? Which parameters control this connection? Is it the same physics as the Rashba-Edelstein effect, where nonequilibrium spin density is induced by a charge current in non-centrosymmetric systems? Is there any other reference connection the Berry dipole with nonequilibrium spin density?

- While reading the manuscript, I could not find a clear connection between the perpendicular tunnelling current and the observed S_z . I had to go back to Sodemann's work and re-derive the nonlinear response based on Eqs. (9) and (10) in order to convince myself that the D_{zz} coefficient is indeed related to the tunnelling current. Therefore, I suggest the authors spend a paragraph explaining the scenario they have in mind and providing the appropriate formulas including the definition of the nonlinear Hall conductivity (Eq. 9 in Ref. 17), and the definition of the Berry curvature dipole (Eq. 10 in Ref. 17). In particular, the reader needs to be able to understand easily that D_{zz} is the Berry curvature dipole that drives the nonlinear Hall effect when measuring a current along x and to the second order of the current along z ...or something like that.

- I think it is important to explicitly state that the nonlinear Hall effect that the authors indirectly probe is different from the one reported in Refs. 21 and 22. In addition, can the authors estimate the D_{zz} coefficient and compare it with the D coefficients extracted from Refs. 21 and 22 for instance?

- Whereas there's very little theory in the manuscript itself, there is a long development in the supplementary materials but focusing on the diffusion equation and completely overlooking the actual band structure of WTe_2 . As I understand, the physics associated with WTe_2 is entirely contained in the D_{zz} coefficient. However, nothing is said about the specific band structure of WTe_2 and how it couples to graphene. In particular, WTe_2 is known for its strong spin-orbit coupling that induces a valley-dependent Zeeman splitting at the K and K' points. In particular, this Zeeman splitting is polarised along z . How does this spin splitting impact the measured current-induced spin density? How can the authors be sure that the MOKE signal they observed is not a mere consequence of this large splitting?

- Does the spin density depend on the orientation of WTe_2 with respect to current injection? In Fig. 1a, the current is injected along the b axis; what happens when it is injected along a ? This could provide an interesting clue to what is going on in this heterostructure.

- Finally, can the authors comment on the physical scale over which the spin density S_z develops? In a traditional spin Hall scenario, S_z vanishes away from the interface over a scale equivalent to the spin diffusion length (See Kato Nature 306, 1910 (2004)); here, I guess the scale is rather governed by the diffusion of the injected current into WTe_2 . Can this scale be resolved by the MOKE measurement?

Point-by-point response to reviewer comments (Manuscript NCOMMS-21-16216-T)

Reviewer #1 (Remarks to the Author):

Comment/Question: This manuscript describes the Kerr microscope measurements with a theoretical explanation of current-induced spin polarization at the heterostructure interface consisting of graphene and van der Waals WTe₂. The results and analyses are intriguing and may warrant publication in Nature Communications, but the presentations and descriptions are not detailed enough to correctly understand the importance and significance of their results. Once the readability problems pointed below are resolved, the manuscript could be acceptable.

Answer: We have made three major changes in order to improve the readability of the manuscript and the presentation of our results therein. Firstly, we have entirely rewritten the introduction with the aim of better placing our work in context to the state-of-the-art. Secondly, we have added the experimental mapping of the local heat dissipation to the main text (manuscript page 3, including new Fig. 2). Thirdly, the entire theory is now described in the new discussion section, for which purpose we have also added further clarifications.

Comment/Question: Firstly, Figures 1a,b, and c, together with captions, are not informative enough to understand the experimental details; the authors should explain how they applied the gate voltage V_g by using an equivalent circuit diagram as in Fig. 3. They should define V_{dc} and R_{2p} and show how their values were experimentally determined in the main text or the method section. The authors should describe it as supplementary information if space does not permit it.

Answer: We have accordingly modified Fig. 1, specifically by labeling the contacts with numbers (in Fig. 1a and b), along with better describing the measurement in the corresponding caption. Furthermore, we employ the same consistent labelling scheme for all other figures. We hope that with these changes the experimental configurations are now unambiguous.

Comment/Question: Secondly, about theoretical description from Eqs. 1 to 3c, it is essential to make the equations more easily understandable for broad readers, the physical meaning of variables such as \square_{ijk} in the equation on line 194, n and τ in Eq. 3 (Why does the Berry curvature density depend on the density n ?).

Answer: In the extended theory section, the relaxation time is now explicitly defined (manuscript page 6, end of first paragraph within left column), and we explain why the Berry curvature density (i.e., anomalous current density) depends on charge density in a way which is somewhat analogous to the relationship between ordinary current density and charge density.

Reviewer #2 (Remarks to the Author):

This work concerns the measurement of a current-induced spin polarization in a WTe₂/Graphene heterostructure and its relation to the Berry curvature. This is a subject of relevance to understand the generation and transport of spins in 2D heterostructures. The measurements as well as the theoretical results appear to be of high quality. However, the main message of the paper is hard to grasp. Moreover, the theoretical proceedings and some of the measurements should be better explained to be accessible to a broad audience.

Answer: We thank the reviewer for the thorough evaluation of our work. As suggested by the reviewer, we have made significant changes in the manuscript in order to better describe the experimental measurements/results, and especially also to improve the presentation of the theoretical model and the underlying mechanism of the current-induced spin polarization. The main changes involve:

(i) We have entirely rewritten the introduction section, in particular to better highlight the major novelty of our work in comparison to the state-of-the-art in the field of 2D spintronics.

(ii) We have moved the section explaining the connection between the Kerr rotation signal, spin accumulation and Berry curvature dipole from the introduction to the new discussion section (manuscript pages 5-6), in order to implement a more coherent and clear presentation of the theory.

(iii) Due to the strong relevance of the current flow distribution in the graphene/WTe₂ heterostructure, we have moved Fig. S2 to the main text (new Fig. 2), and added a section to explain the maps in terms of local heat dissipation (manuscript page 3).

(iv) We have added a schematic illustration to former Fig. 4 (now Fig. 5, panel a), with the aim of conveying the connection between the electrical current direction, spin polarisation and the electric field of the incoming light.

Comment/Question: The main experimental point is the demonstration that an in-plane current flowing in graphene induces a (strong?) spin accumulation in a WTe₂ layer on top of graphene, as measured by the magnetooptic Kerr effect. The spin polarization is out-of-plane and depends on the type of charge carriers dominating transport in graphene. This observation is attributed to a diffusive out-of-plane current flowing from graphene to WTe₂ and to the nonzero Berry curvature dipole of the non-centrosymmetric WTe₂ layer. Furthermore, the Kerr effect is related to the nonlinear Hall effect of WTe₂.

After reading the paper, one still does not understand if the Kerr effect measurements provide a quantitative measurement of the Berry curvature dipole, nonlinear Hall conductivity, and spin accumulation in WTe₂. Whereas it can be argued that the Kerr angle provides spatially-resolved

signatures of these effects, their quantitative evaluation relies on so many assumptions and parameters that only order of magnitude estimates can be given. Indeed, the fits according to Eq. S18 are used only to confirm the qualitative agreement of the model with the experimental results. The authors should clarify what the scope and limitations of the measurements are.

Answer: We note that, in principle, Kerr measurements indeed enable a quantitative measurement of the Berry curvature dipole via the gyrotropic Hall effect, as theoretically established by König et al. [Phys. Rev. B 99, 155404 (2019)], which in turn is closely related to the nonlinear Hall effect, as demonstrated by Sodemann and Fu [PRL 115 (2015), 216806]. To connect the Berry curvature dipole, which is a Fermi surface property in the low temperature limit, to the Kerr angle, a proper description of the optical transition matrix elements is required. As one example, Son et al. [PRL 123 (2019), 036806] have used the Kerr effect to experimentally determine the strain-induced Berry curvature dipole via the current-induced magnetization in semiconducting MoS₂, employing a two-band model for a resonant interband excitation.

However, a quantitative comparison between experiment and theory is challenging for the following two reasons. First, non-linear Hall responses and their close cousin, the circular photogalvanic effect, see Deyo et al. [arXiv:0904.1917] and de Juan et al. [Nat. Comm. 8 (2017), 1], depend on a non-universal scattering time. Second, the theoretical treatment of interband transitions at optical frequencies of semimetallic (or gapless) systems, such as WTe₂, is more challenging, because often many possible interband transitions exist.

In our theory we exploit that, for effective two-band models, the Berry curvature dipole description is valid even at optical frequencies connecting valence and conduction bands (the frequency dependence only enters as a prefactor [see Eq. (18) of Phys. Rev. B 99, 155404 (2019)]). Towards accessing the spatial distribution of the Kerr response, we employ topological diffusion theory, which, however, is strictly valid only in a one-band model (see also the works by Deyo et al., and Sodemann/Fu, cited above). As we know that the optical frequency enters the homogeneous Kerr response only as a prefactor, it is expected that the inhomogeneous result of the topological Fick theory is applicable at the very least for effective two-band models.

It should be noted that at optical frequencies, provided that the crystal is sufficiently clean and the electronic bands undisturbed, mechanisms related to disorder scattering (i.e., extrinsic contributions) are not effective, but rather the intrinsic Berry curvature related mechanisms are dominant. From an experimental point of view, a THz to mid-IR read-out could in principle be used for a quasi-resonant excitation, however, at the cost of sacrificing spatial resolution for the case of a far-field read-out.

To clarify the scope and limitations of our approach, we have added a paragraph to manuscript page 7 (left column), which summarizes the above arguments.

Comment/Question: The generation of an out-of-plane spin accumulation can in principle be assigned to different mechanisms depending on the actual direction of current flow in the heterostructure. Also, the spin texture of graphene in proximity to WTe₂ is known to acquire out-of-plane components. The authors should explain more clearly if and why the out-of-plane spin polarization is an exclusive signature of the nonlinear Hall effect. They should also specify more clearly how the WTe₂ is electrically connected for the experiments reported in Fig. 1.

Answer: One possible alternative mechanism of current-induced spin polarization involves the spin Hall effect in WTe₂, which, however, neither for lateral nor vertical current flow can account for the observed signals of opposite signs at the left and right edge of the junction (see Fig. 1a). This is due to the fact that this effect requires the directions of electrical current, spin current and spin to be mutually orthogonal.

A second option is current-induced spin polarization through lateral charge transport through the proximitized graphene. While the associated spin texture may indeed have out-of-plane spin components, this scenario would be expected to lead to a Kerr signal that is homogeneously distributed along the graphene/WTe₂ heterointerface, rather than the appearance of local signals at the graphene/WTe₂ edges as observed in our experiments.

We have added short explanations on why these two effects can be ruled out to the beginning of the (new) discussion section (manuscript pages 4/5).

Furthermore, with regard to the electrical measurement configuration, we have labeled the contacts in Fig. 1a and b, and specify their use in the corresponding caption. Thus, it now becomes evident that for the Kerr rotation measurements the WTe₂ contacts are floating, i.e. they draw no electrical current.

Comment/Question: Is there any relationship between the laser polarization, crystal axes of WTe₂, and the spin-polarization direction? Changing the orientation of the WTe₂ layer relative to the current flow and/or the linear polarization direction of the laser could lend further support to the authors' analysis.

Answer: We have measured the angle-dependence of the first- and second-harmonic KR signal for a graphene/WTe₂ heterostructure and a bare WTe₂ ribbon, as shown in the Figure below.

The second harmonic signal detected on the WTe₂ ribbon - featuring a cosinus-like modulation (panel c) - can be attributed to a birefringence effect [Sonowal et al., Phys. Rev. B. 100 (2019), 085436]. The latter manifests itself also in the second harmonic signal of the heterostructure (panel a), although there is a noticeable difference between the two measurement positions, which may well arise from symmetry breaking due to the attached graphene.

It should be mentioned that, owing to the dominant contribution of the birefringence effect, the sign changes visible in panels a-c do not imply an inversion of the spin direction.

Unfortunately, at the present stage we do not have a suitable model to consistently explain the angle dependence of the first harmonic signal observed for the heterostructure (panel b), and hence cannot use this result to further consolidate our Berry curvature-based mechanism of current-induced spin polarization. For that reason, we prefer to not add the new data to the manuscript.

Comment/Question: The discussion of the nonlinear Hall conductivity is not straightforward. The nonlinear Hall conductivity is related to the product of two electric field components, which in transport measurements reduce to the square of the electric field driving the current (see, e.g., ref. 22). However, in this case there is no electric field applied in the z-direction, only the diffusion current arising from the carriers' concentration gradients. Perhaps the authors imply that the electric field E_j is the one that builds up as a result of the diffusion process, but this is not explained. Moreover, the second component of the electric field is the one due to the incident light at frequency ν . The interplay of the different types of electric fields in determining the Hall conductivity is not at all evident to readers who have no notion of optical measurements. The nonlinear Hall effect and the different types of electric fields should be introduced in a more pedagogical way. Presenting a schematic diagram showing the different field/current directions and the spin polarization direction allowed by symmetry would help.

Answer: Indeed, the relevant electric fields in z-direction, which generate the nonlinear Hall conductivity, are linked to the diffusion process in WTe₂.

To clarify the interplay of the relevant electric fields at different frequencies, we have added a new schematic as Fig. 5a, which illustrates the optical field for readout of the local Hall conductivity, as well as the slow quasi-DC fields related to the local diffusion currents. Furthermore, we have restructured the description in the text (manuscript page 6, left column). We hope that with these changes our approach is described in a more pedagogical and accessible way.

Comment/Question: Moreover, the authors should discuss how the Kerr rotation depends on the current amplitude in graphene (data in Fig. S3a) and on the laser intensity in order to support their model.

Answer: We have made two additions to address this point.

Firstly, with respect to the dependence of Kerr rotation on the current amplitude, we have added a sentence on manuscript page 6 (left column), where we comment on the dependencies which are now shown in Figure S2a. In short, the first harmonic Kerr rotation signal displays an almost linear dependence above the bias voltage threshold (where the Fermi level reaches the Dirac point of the proximitized graphene), while the second harmonic signal shows a close-to-quadratic bias dependence as expected for heat dissipation.

Secondly, regarding the laser intensity dependence, we have added a new figure to the Supplementary Information (Fig. S5) which shows that the Kerr angle is independent of the laser intensity, in accordance with our model of a current induced Kerr effect, where the optical excitation serves only as a read-out. This is distinct to possible higher order effects, where the optical excitation modifies directly the Kerr response [<https://arxiv.org/abs/2103.08173>]. At the same time, the fact that the Kerr angle is independent of the strength of the probe field excludes local photocurrents to be the origin of the Kerr response and corresponding spin accumulation. We have added a short, corresponding explanation to the main text, directly after the statement on the current dependence (manuscript page 6, left column).

Comment/Question: The discussion also lacks a broader perspective relating these qualitative findings to other measurements of charge-spin interconversion in 2D heterostructures. What do these results add to the experiments revealing the spin Hall effect and inverse spin galvanic effect in graphene/transition metal dichalcogenide systems?

Answer: We have changed the introduction section and added a new paragraph (manuscript page 7, left column) in order to put our findings in perspective to previous studies of charge-to-spin conversion in graphene/TMDC heterostructures, and to clarify the scope and possible opportunities of our approach.

In short, from our point of view, our study enables us to “optically read-out the local spin polarisation (or Hall conductance), and correlate the signals with the independently determined

charge current flow through the heterostructure". This local readout is distinct from the information that can be gained from global charge transport measurements.

Our optical experiments would in principle allow accessing the Berry curvature dipole, albeit this requires that the interaction at optical frequencies is sufficiently well described. Therefore, in the modified discussion section (manuscript page 7), we discuss the limitations of the present model as well as possible improvements, such as resonant two-band excitation in the THz regime.

Comment/Question: Aside from these points, I suggest to add a top view of the wiring diagrams for both Fig. 1 and 2 and a clearer explanation of what is actually measured. What is R_{2p} in Fig. 1c, a two-point resistance vs V_g ? The tunnelling spectroscopy is $\delta V / \delta I$; does δV measure an ac voltage and $V_{ds,meas}$ a dc voltage?

Answer: We have introduced a consistent labelling scheme for the contacts and the measured quantities throughout the whole manuscript, including the relevant figures. We hope that with these changes the experimental configurations become clear and unambiguous.

Comment/Question: In the Supplementary Information, the theoretical section is hard to follow, at least for a person that does not share the same background as the one who wrote it. I would first expect a discussion of how the nonlinear Hall effect emerges from the topological Fick theory. This would allow the reader to identify at first sight which quantities need to be modelled in order to understand the measurements. Instead, the nonlinear Hall conductivity is given almost for granted on line 138 based on results presented in Refs. 17 and 32. Moreover, the role of the imbalance current could be introduced in a more intuitive way. Finally, in connecting the Kerr rotation to the Hall conductivity (Eq. S17), the Berry curvature dipole is replaced by the Berry curvature Ω and then dropped, which brings back the question of how useful are the measurements to determine topological parameters. Several quantities that appear in this equation have not been defined (v_W , v_w , W , I , ...). p_F is likely the Fermi momentum in units of $1/\text{length}$, but this is not written. Overall, the notation is rather involved and sometimes confusing. For example, the Berry curvature dipole and diffusion constants share the same symbol D .

Answer: Following this suggestion and the request of ref. 3 (see below), the theory section in the Supplementary Information now starts with a review of the Kerr response. Moreover, as a conceptual counterpart, the section on topological Fick theory now clearly expresses the Kerr angle (Hall response) in terms of the Berry curvature density. We also adjusted the section on the imbalance current in a way, which we believe is more pedagogical.

Regarding the estimate of the strength of the effect: Admittedly, replacing the Berry curvature dipole by typical values of the Berry curvature allows us only to roughly estimate its magnitude. It is not meant to be a microscopic derivation of the experimentally measured observable;

nonetheless, we believe our theory provides a valuable theoretical framework for future, detailed first-principle calculations which could yield a quantitative answer.

We have clarified the use of symbols. We agree that the notation of 'D' for both Berry curvature dipole and diffusion constant is somewhat unfortunate, but since both notations appeared in the literature before, we feel that we should stick to standard notation, and use the superscript 'Omega' to distinguish them.

Reviewer #3 (Remarks to the Author):

In this work, Powalla et al. report the measurement of the current-induced spin density at the interface between WTe_2 and graphene monolayer probed by magneto optical Kerr microscopy. The authors observe a nonequilibrium spin density polarised normal to the plane of the heterostructure, that displays an antisymmetric profile along the injection direction, and whose magnitude can be controlled by a gate voltage. The authors interpret their experimental results as a consequence of the Berry curvature dipole experienced by the carriers tunnelling from graphene to WTe_2 .

Comment/Question: This is an interesting result that will certainly attract significant attention in the community. An interesting idea is the connection between Berry curvature dipole, nonlinear anomalous Hall effect and nonequilibrium spin density. However, whereas the qualitative connection between S_z and the nonlinear Hall effect and its associated Berry dipole D_{zz} is rather reasonable, I find the formal relationship is still not well established. The manuscript would gain in clarity by further developing this connection.

As a matter of fact, WTe_2 is known to display nonlinear anomalous Hall effect, which has been theoretically associated with the Berry curvature dipole. In Eq. (1), the authors establish a qualitative connection with the S_z component of the spin density and the Berry curvature dipole. It would be more convincing to provide the explicit expression that relates these two quantities rather than only providing (unjustified) proportionality relations. What is the physics connected S_z and the nonlinear Hall effect? Which parameters control this connection? Is it the same physics as the Rashba-Edelstein effect, where nonequilibrium spin density is induced by a charge current in non-centrosymmetric systems? Is there any other reference connection the Berry dipole with nonequilibrium spin density?

Answer: We have substantially extended the introductory part of the theory section in the Supplementary Information to properly explain the formal connection between Berry curvature dipole, nonlinear Hall effect, and non-equilibrium spin density in quite some detail. Indeed, the relationship between spin-density and Kerr angle is, to a certain extent, phenomenological, while the relationship between Kerr angle and Berry curvature dipole is microscopically much more precise. Yet, using symmetry arguments, we demonstrate that despite this phenomenological nature, all proportionalities are well defined.

We have added one sentence to the main text (a few lines after equation (1)) in order to refer the reader to the Supplementary Information in this context.

Comment/Question: While reading the manuscript, I could not find a clear connection between the perpendicular tunnelling current and the observed Sz. I had to go back to Sodemann's work and re-derive the nonlinear response based on Eqs. (9) and (10) in order to convince myself that the Dzz coefficient is indeed related to the tunnelling current. Therefore, I suggest the authors spend a paragraph explaining the scenario they have in mind and providing the appropriate formulas including the definition of the nonlinear Hall conductivity (Eq. 9 in Ref. 17), and the definition of the Berry curvature dipole (Eq. 10 in Ref. 17).

In particular, the reader needs to be able to understand easily that Dzz is the Berry curvature dipole that drives the nonlinear Hall effect when measuring a current along x and to the second order of the current along z...or something like that.

Answer: We have restructured the discussion of the current-induced Hall response (in the new discussion section). In particular, we now explicitly mention that λ_{zz} (and therefore D_{zz}) of the heterostructure is the relevant quantity in our experimental configuration. Moreover, we have added a new Figure 5a which illustrates the relevant currents, electric fields and their directions in our experiment.

Comment/Question: I think it is important to explicitly state that the nonlinear Hall effect that the authors indirectly probe is different from the one reported in Refs. 21 and 22. In addition, can the authors estimate the Dzz coefficient and compare it with the D coefficients extracted from Refs. 21 and 22 for instance?

Answer: Our results are indeed distinct to the previously reported nonlinear anomalous Hall effect in WTe₂. In the 2D limit of WTe₂, i.e. for few layer samples, the crystal symmetry is reduced to Pm compared to Pm2₁ for bulk WTe₂. Only this symmetry breaking allows the observation of the in-plane anomalous Hall effect, as reported by Kang et al. [Nature Mater. 18 (2019), 324] and Ma et al. [Nature 565 (2019), 337], and predicted by Sodeman and Fu [Phys. Rev. Lett. 115 (2015), 216806]. In a simple picture, this in-plane nonlinear anomalous Hall effect goes along with an out-of-plane magnetization as well, since the applied current, the induced Hall current and the induced spin polarization are all required to be perpendicular to each other.

In comparison, as our samples consist of (quasi-)bulk WTe₂, we do not detect an out-of-plane spin polarization for currents applied exclusively along the in-plane axes of WTe₂ (see Supplementary Figure S7). Rather, the graphene/WTe₂ heterointerface gives rise to a symmetry-allowed collinear response, where the directions of current density and spin density are aligned parallel or antiparallel due to the Dzz term. We would like to emphasize that in 2D crystals, such as few-layer WTe₂, the Berry curvature dipole behaves as a pseudo-vector $D = (D_x, D_y)$ contained in the 2D plane, and therefore it is not meaningful to compare it to our values of Dzz, as was suggested by the reviewer.

In order to convey the difference between the reported anomalous Hall effect and our present results, we have added two sentences to manuscript page 2 (right column, end of first paragraph).

It is furthermore noteworthy that for bulk WTe₂, a recent study has established a large out-of-plane nonlinear Hall effect along the c-axis, which is symmetry-allowed for currents applied along the in-plane a- and b-axes [Tiwari et al., Nature Commun. 12 (2021), 2049]. In their experiment, the induced magnetization would be required to be only in-plane, by symmetry considerations, which is again distinct to our observations. In addition, the underlying microscopic mechanism of this c-axis non-linear Hall effect was shown to be dominated by extrinsic effects rather than the intrinsic Berry curvature dipole.

Comment/Question: Whereas there's very little theory in the manuscript itself, there is a long development in the supplementary materials but focusing on the diffusion equation and completely overlooking the actual band structure of WTe₂. As I understand, the physics associated with WTe₂ is entirely contained in the D_{zz} coefficient. However, nothing is said about the specific band structure of WTe₂ and how it couples to graphene. In particular, WTe₂ is known for its strong spin-orbit coupling that induces a valley-dependent Zeeman splitting at the K and K' points. In particular, this Zeeman splitting is polarised along z. How does this spin splitting impacts the measured current-induced spin density? How can the authors be sure that the MOKE signal they observed is not a mere consequence of this large splitting?

Answer: We agree that in the model that we have developed, the microscopic details of the bands governing the current-induced Kerr rotation are fully encoded by the structure of the Berry curvature and the corresponding structure of the Berry curvature dipole.

In the low temperature limit, the nonlinear Hall effect and correspondingly also the current-induced Kerr rotation are properties of the Fermi surface, i.e., only states close to the Fermi level contribute, and indeed the Berry curvature dipole captures the relevant properties of the Fermi surface.

On this basis, we also exclude the spin texture of the proximitized graphene to be the origin of the observed current-induced Kerr effect. Specifically, the band structure of graphene proximitized by WTe₂ is expected to show (i) a gap opening of several meV at the Dirac point, (ii) an in-plane spin component due to a Rashba-like interaction, and (iii) an out-of-plane spin texture due to a valley-Zeeman like interaction. Generally, such spin textures are expected to be most pronounced near the opened gap at the Dirac point, while they decay quickly with increasing the Fermi level, and they are opposite for electrons and holes. Thus, for the graphene/WTe₂ heterostructure, the Kerr effect induced by an out-of-plane current should be maximized (peak-like) for the Fermi level near the gap, in contrast to our experimental observation of an onset (step-like) of the Kerr effect, when the Fermi level is tuned to the Dirac point of the proximitized graphene.

While it would definitely be very interesting to see effects from the hybridized interfacial spin texture, our experiments and the above arguments indicate that the local out-of-plane currents

are the dominating effect. Here, the role of the interface is to break the overall symmetry and to enable the interlayer carrier exchange. One option to access effects due to the proximitized graphene may be experiments on devices that operate closer to the 2D limit, which could be implemented by combining very thin WTe₂ (2-3 layers) with graphene.

Comment/Question: Does the spin density depend on the orientation of WTe₂ with respect to current injection? In Fig. 1a, the current is injected along the b axis; what happens when it is injected along a? This could provide an interesting clue to what is going on in this heterostructure.

Answer: We did not experimentally investigate the question of what happens in case of current injection along the a-axis of WTe₂ in graphene/WTe₂ heterostructures, as this would require a rather big device fabrication effort (including an etching process), and the gained additional information may well be limited. This is because in principle, we would still expect similar Kerr signals at the junction edges (as in Fig. 1a), as the out-of-plane spin polarisation arises from the electrical current in z-direction (i.e., along the c-axis in WTe₂, see also previous answer), such that the actual in-plane axis direction is most likely not relevant due to symmetry reasons.

The above assumption gains support by additional control experiments that we have performed on bare bulk WTe₂, with current being injected along the a- vs. b-axes (see new Fig. S7). Along both the a-axis and b-axis, we do not see a Kerr response from a pure in-plane current in bulk WTe₂, as expected based upon symmetry considerations of bulk WTe₂.

Comment/Question: Finally, can the authors comment on the physical scale over which the spin density S_z develops? In a traditional spin Hall scenario, S_z vanishes away from the interface over a scale equivalent to the spin diffusion length (see Kato Nature 306, 1910 (2004)); here, I guess the scale is rather governed by the diffusion of the injected current into WTe₂. Can this scale be resolved by the MOKE measurement?

Answer: This is a very good point. As can be seen in the figure below, along the direction of the graphene the Kerr rotation signal (blue line, panel a) is localized at the heterointerface with an extent consistent with the optical resolution (orange line, panel a). By contrast, along the direction of WTe₂, the Kerr map clearly displays a signal which extends away from the heterostructure interface (blue line, panel b) much larger than the optical resolution (orange line, panel b). The resolved length scale is about 2 μm (shaded area, panel b), where we have already considered the spatial resolution of our experiment determined from the reflectance image.

The observed length scale in the WTe₂ is consistent with a recent study on MoTe₂, where an anomalously long spin diffusion length of 2 μm was found, even at room temperature. However, for the sake of completeness we mention that for WTe₂ itself, Zhao et al. [Phys. Rev. Research 2 (2020), 013286] determined a spin diffusion length of only 8 nm, which clearly could not be resolved by our far-field measurement. We note that Zhao et al. had to assume a spin Hall angle of 0.013 [as determined in Nature Physics 13 (2017), 300], since the spin Hall angle and spin diffusion length could not be determined independently in their inverse spin Hall effect measurement.

At the present stage, we are unable to decide whether it is realistic that the spin diffusion length in WTe₂ is approximately three orders of magnitude lower than in MoTe₂, and how this difference may be explained. In any case, we would like to maintain that full clarification of this issue would go beyond the scope of our current study.

Reviewers' Comments:

Reviewer #1:

Remarks to the Author:

The readability of the manuscript is significantly improved. As mentioned in the previous report, the results and analyses are intriguing. They warrant publication in Nature Communications.

Reviewer #2:

Remarks to the Author:

The manuscript has been substantially improved, in particular the description of the measurements and the introduction of the theoretical sections. One point that remains open, however, is the angular dependence of the Kerr rotation reported in the rebuttal letter, which has been measured by the authors in response to a previous comment. I am particularly worried by the fact that the first harmonic Kerr signal has a very similar angular dependence to the second harmonic Kerr signal, which is mainly attributed to birefringence effects. I wonder if there is a connection between the two and whether this challenges the main conclusion of the paper, namely that the first harmonic Kerr signal originates from the Berry curvature dipole. I am also worried by the fact that the authors do not wish to include these data in the manuscript because they are not understood.

As mentioned on page 16 of the SI, the current-induced magnetization S_z can be of spin or orbital origin. Can the authors give arguments in support of one or the other? In the end, one wonders if the Kerr rotation can provide quantitative information on relevant physical parameters of the graphene/WTe₂ heterostructure or only a qualitative signature of different effects.

Finally, I miss a comprehensive summary of the main results accessible to a mid-IQ reader (or reviewer). Neither the abstract, introduction, or conclusions report a specific description of the main findings. Something similar or better to the second paragraph of my previous report or the first paragraph of the report of Referee #3.

Reviewer #3:

Remarks to the Author:

The authors have provided detailed responses to the referees' comments and made substantial modifications to the manuscript. I believe it did gain in clarity. The experimental results are intriguing and the physical interpretation brings interesting ideas that will need to be confirmed. I have no further questions or comments.

Reviewers' Comments:

Reviewer #1:

Remarks to the Author:

The readability of the manuscript is significantly improved. As mentioned in the previous report, the results and analyses are intriguing. They warrant publication in Nature Communications.

Reviewer #2:

Remarks to the Author:

The manuscript has been substantially improved, in particular the description of the measurements and the introduction of the theoretical sections. One point that remains open, however, is the angular dependence of the Kerr rotation reported in the rebuttal letter, which has been measured by the authors in response to a previous comment. I am particularly worried by the fact that the first harmonic Kerr signal has a very similar angular dependence to the second harmonic Kerr signal, which is mainly attributed to birefringence effects. I wonder if there is a connection between the two and whether this challenges the main conclusion of the paper, namely that the first harmonic Kerr signal originates from the Berry curvature dipole. I am also worried by the fact that the authors do not wish to include these data in the manuscript because they are not understood.

As mentioned on page 16 of the SI, the current-induced magnetization S_z can be of spin or orbital origin. Can the authors give arguments in support of one or the other? In the end, one wonders if the Kerr rotation can provide quantitative information on relevant physical parameters of the graphene/WTe₂ heterostructure or only a qualitative signature of different effects.

Finally, I miss a comprehensive summary of the main results accessible to a mid-IQ reader (or reviewer). Neither the abstract, introduction, or conclusions report a specific description of the main findings. Something similar or better to the second paragraph of my previous report or the first paragraph of the report of Referee #3.

Reviewer #3:

Remarks to the Author:

The authors have provided detailed responses to the referees' comments and made substantial modifications to the manuscript. I believe it did gain in clarity. The experimental results are intriguing and the physical interpretation brings interesting ideas that will need to be confirmed. I have no further questions or comments.

Reviewer #1 (Remarks to the Author):

The readability of the manuscript is significantly improved. As mentioned in the previous report, the results and analyses are intriguing. They warrant publication in Nature Communications.

Answer/Comment:

We thank the reviewer for the positive assessment and her/his efforts to carefully review our manuscript.

Reviewer #2 (Remarks to the Author):

The manuscript has been substantially improved, in particular the description of the measurements and the introduction of the theoretical sections. One point that remains open, however, is the angular dependence of the Kerr rotation reported in the rebuttal letter, which has been measured by the authors in response to a previous comment. I am particularly worried by the fact that the first harmonic Kerr signal has a very similar angular dependence to the second harmonic Kerr signal, which is mainly attributed to birefringence effects. I wonder if there is a connection between the two and whether this challenges the main conclusion of the paper, namely that the first harmonic Kerr signal originates from the Berry curvature dipole. I am also worried by the fact that the authors do not wish to include these data in the manuscript because they are not understood.

Answer/Comment:

We agree with the reviewer that the experimentally observed angular dependence of the Kerr signal requires an explanation. The most important question is to distinguish heating effects in birefringent materials from spin accumulation associated with the Berry curvature dipole.

A possible avenue to further separate thermal effects from spin effects could be to exploit their characteristic time scales resulting in different frequency dependencies with respect to the current modulation. The limiting time constant of the thermal signal is expected to be cooling to the substrate (ns – ms), whereas for the spin signal it is expected to be related to the electronic and spin dynamics of the carriers in WTe₂ (ps - ns). We have performed such additional experiments. Unfortunately, the bandwidth of our measurement (Hz – 100 kHz) was not sufficient to see a significant suppression of the second-harmonic thermal signal, because the thermal conductivity of the underlying Si/SiO₂ substrate is too effective in our samples. This hurdle may in principle be overcome by fabricating, e.g., suspended, gateable structures on thin membranes, however, this would be quite time consuming and goes beyond the scope of the current work.

Nonetheless, the difference between the angular dependency observed for the WTe₂ ribbon and for the graphene/WTe₂ heterostructure (which we showed already in the first rebuttal letter) yields a valuable clue regarding the respective contributions of the two mechanisms. As we explain in detail in the extended Supplementary Information (page 16), birefringence effects in a non-magnetic material generally lead to a zero first harmonic signal, while the second harmonic signal averages to zero upon integration over the polarization angle (as we have observed for the WTe₂ ribbon; see left plot below). In contrast, for the graphene/WTe₂ heterostructure we observe a finite angular average of both, the first and second harmonic signal (the former is shown in the right plot below). Following standard literature, this behavior can only be attributed to magnetization, which in our case is current-induced through the Berry-curvature dipole effects. Accordingly, we concentrate in the main text on the effect of time-reversal symmetry breaking.

As suggested by the reviewer, we have included the angular dependence of the first and second harmonic signal, specifically in the Supplementary Information (as Fig. S8). Moreover, in order to provide a complete picture of the possible mechanisms that may be involved in the Kerr signal generation, we have added a short section to the main text (page 6), wherein we mention that a contribution of heating effects cannot be entirely ruled out.

As mentioned on page 16 of the SI, the current-induced magnetization S_z can be of spin or orbital origin. Can the authors give arguments in support of one or the other? In the end, one wonders if the Kerr rotation can provide quantitative information on relevant physical parameters of the graphene/WTe₂ heterostructure or only a qualitative signature of different effects.

Answer/Comment:

As the reviewer correctly observes, based on symmetry, it is not possible to distinguish current-induced spin polarization from orbital magnetization – in particular given the importance of spin-orbit coupling. However, in typical materials, the orbital magnetization is typically only a few percent of μ_B and thus small [see, e.g., the review by T. Thonhauser, "Theory of Orbital Magnetization in Solids"; *Int. J. Mod. Phys. B.* 25 (2011), 1429]. In this context, one should keep in mind that recent proposals of massive orbital magnetization in twisted bilayer graphene are really exceptional and an anomalous characteristic of twistronic flat bands [see, e.g., W.-Y. He et al., *Nature Communications*, 11 (2020), 1650]. We emphasize again, that the Berry-curvature dipole calculation/interpretation is independent of the potential ambiguity between magnetization stemming from spin or orbital origin.

Furthermore, with respect to the possibility to extract quantitative information from the Kerr rotation angle, realistically this would be quite difficult, as we explain in lines 395 to 420 of the manuscript. In short, determining a reliable value for the Berry curvature dipole would require detailed knowledge of the optical conductivities and the carrier scattering times. In the specific case of WTe₂, (experimentally) accessing the former quantity is complicated by the simultaneous presence of several interband transitions.

Finally, I miss a comprehensive summary of the main results accessible to a mid-IQ reader (or reviewer). Neither the abstract, introduction, or conclusions report a specific description of the main findings. Something similar or better to the second paragraph of my previous report or the first paragraph of the report of Referee #3.

Answer/Comment:

We have modified and expanded the conclusion section of the manuscript (page 7), in order to better highlight and summarize the major achievements of our work.

Reviewer #3 (Remarks to the Author):

The authors have provided detailed responses to the referees' comments and made substantial modifications to the manuscript. I believe it did gain in clarity. The experimental results are intriguing and the physical interpretation brings interesting ideas that will need to be confirmed. I have no further questions or comments.

Answer/Comment:

We thank the reviewer for the valuable input, and her/his efforts to carefully review our manuscript.

Reviewers' Comments:

Reviewer #2:

Remarks to the Author:

The authors have appropriately addressed my last comments. The angular dependence of the first and second harmonic Kerr signals is now explicitly discussed. The manuscript has gained in clarity. The results shed light on the mechanisms that govern generation of spin currents and the nonlinear Hall effect in 2D heterostructures and I am happy to recommend it for publication.

Point-by-point response to the reviewer comments

Reviewer #2 (Remarks to the Author):

The authors have appropriately addressed my last comments. The angular dependence of the first and second harmonic Kerr signals is now explicitly discussed. The manuscript has gained in clarity. The results shed light on the mechanisms that govern generation of spin currents and the nonlinear Hall effect in 2D heterostructures and I am happy to recommend it for publication.

Response:

We are happy that reviewer #2 has stated that we have appropriately addressed his/her last comments, and that publication of our manuscript is recommended. We thank the reviewer for all his/her efforts.